# Energy Efficiency of Oat:Pea Intercrops Affected by Sowing Ratio and Nitrogen Fertilization

Gerhard Moitzi [1,*], Helmut Wagentristl [1], Hans-Peter Kaul [2], Jaroslav Bernas [3] and Reinhard W. Neugschwandtner [2]

1.  Experimental Farm Groß-Enzersdorf, Department of Crop Sciences, University of Natural Resources and Life Sciences (BOKU), Schloßhofer Straße 31, 2301 Groß-Enzersdorf, Vienna, Austria
2.  Institute of Agronomy, Department of Crop Sciences, University of Natural Resources and Life Sciences (BOKU), Konrad-Lorenz-Straße 24, 3430 Tulln an der Donau, Vienna, Austria
3.  Department of Agroecosystems, Faculty of Agriculture and Technology, University of South Bohemia, Branišovská 1457, 370 05 České Budějovice, Czech Republic
*   Correspondence: gerhard.moitzi@boku.ac.at

**Abstract:** This study analyzed energy input (direct and indirect), energy output, net-energy output, energy use efficiency, energy intensity, and the energy productivity of oat:pea intercrops as affected by sowing ratio (oat:pea (%:%): 100:0, 75:25, 50:50, 25:75, 0:100) and nitrogen (N) fertilization (0, 60, 120 kg N ha$^{-1}$). The two year field experiment was conducted on a calcaric Chernozem soil in the north-western part of the Pannonian Basin. The results for grain yield showed that pure stands of oat and pea had a higher energy use efficiency and energy intensity than intercrops, indicating that pure stands used the growing factors more efficiently than intercrops. The energy use efficiency was higher in pure pea than pure oat. The energy productivity for the above-ground biomass production was much more affected by the factor N fertilization than by the factor sowing ratio. The highest energy productivity of grain N yield and above-ground biomass N yield was achieved in pure pea stands (0:100). N in plant residues of the zero N fertilization variant required 68% lower technical energy than N from mineral fertilizer. The sowing rate of the intercrops is a management tool to trade-off between the benefits of the in-field biodiversity and energy efficiency.

**Keywords:** intercrops; N fertilization; sowing rate; energy use efficiency; energy intensity; energy productivity; Pannonian climate

## 1. Introduction

Cereal-legume intercrops have the potential, compared to monocrops, to use limiting growth resources more effectively, reduce pest incidence, have higher protein yields, and improve soil fertility through biological dinitrogen fixation (N$_{FIX}$) [1]. For example, the intercropping of fenugreek and buckwheat resulted, compared to their corresponding sole crops, in a higher biomass and seed yield, mainly to the better performance of buckwheat in intercrops, in higher nitrogen (N) plant concentrations and uptake, as well as in increased applied N use efficiency and applied N recovery efficiency. Yields and nutrient uptake were more enhanced by broiler litter compared to chemical fertilization [2,3]. The intercropping of pea with linseed could improve the root length density and root dry matter of pea [4]. Intercrops are extensively grown in traditional, labor-intensive, small-scale farming systems in tropical countries where their advantages will be higher than in temperate regions [5]. There is also an increasing interest in intercropping in highly mechanized agriculture systems in temperate regions [6], where they can contribute to sustainable intensification, increasing productivity, yield stability and ecosystem services [7]. Although intercropping fits best for organic farming, it might also be suitable in conventional cropping systems [8]. In conventional cropping systems in temperate regions, mineral N is used to achieve high yields [9]. Several intercropping systems have been tested recently in Pannonian climate

conditions in Eastern Austria. Autumn-sown intercrops of wheat under rapeseed had an overall higher productivity compared with sole-cropping of wheat and rapeseed, with rapeseed being the dominant crop. The total cropping system performance increased with adding N fertilizer, with rapeseed reacting stronger to N fertilization in intercrops than wheat [10].

A comprehensive study on the spring-sown intercrops of oat and pea also under Pannonian climate conditions in Eastern Austria focusing on yield, yield components, the concentrations and uptake of macro- and micro-nutrient, nitrogen fixation, and environmental impacts, have reported either advantages or disadvantages of intercropping regarding the parameters observed. Oat was the dominant partner in the intercrops with an increasing competitive ability with higher N fertilization. Land equivalent ratio analysis showed that oat:pea intercrops attained higher residues yields compared to pure stands but failed to achieve a higher grain yields as the harvest indices were impaired. However, as the grain N concentrations of oat were higher in intercrops, intercrops could attain higher grain N yields in unfertilized treatments compared to pure crop stands. Consequently, oat:pea intercrops can be for reasonable for producing grain feed at a low N input level [11,12]. Additionally, both considered intercropping ratio and N fertilization affected concentrations and yields of some macro- and micronutrients in the grain and especially in residues, making intercropping also a reasonable strategy especially when residues are also used for ruminant feeding [13,14]. Intercropping considerably reduced $N_{FIX}$ compared to pure pea in oat:pea intercrops. Neither the cropping ratio nor a low amount of N fertilization affected the $N_{FIX}$ per unit dry matter of pea [15]. Assessing the environmental impacts by life cycle assessment with a focus on the grain N yield showed that oat:pea intercrops could result in lower environmental impacts and that fertilizer inputs did not necessarily cause the highest environmental impacts as an appropriate grain N yield must be reached to balance the environmental impacts resulting from the fertilizer inputs [16].

To further contribute to the comprehensive understanding of oat:pea intercrops, this study aimed to conduct an energy efficiency analysis of oat:pea intercrops affected by sowing ratio and N fertilization with a focus on: (a) energy input (direct and indirect); (b) energy output; (c) net-energy output; (d) energy use efficiency; (e) energy intensity; and (f) energy productivity. Energy intensity and energy productivity were calculated for crop yields and N yields.

## 2. Materials and Methods

### 2.1. Experimental Site and Climatic Conditions

The experiment was conducted at the Experimental Station of the University of Natural Resources and Life Sciences (BOKU), Vienna in Raasdorf (48°14′ N, 16°33′ E; 153 m a.s.l.) in 2010 and 2011. The silt loam soil (pHCaCl$_2$: 7.6 and soil organic carbon: 23 g kg$^{-1}$) is classified as a calcaric Chernozem of alluvial origin [17] (WRB, 2006). The field site is located in the east of Vienna (Austria) on the edge of the Marchfeld plain, which is an important crop production region in the north-western part of the Pannonian Basin. The Pannonian climate area is characterized by hot summers with low rainfall and cold winters with little snow. The long-term (1980–2009) mean values for annual temperature and annual precipitation were 10.6 °C and 538 mm, respectively. The temperature, compared to the long-term average, was considerably higher in 2010 in June and July and in 2011 in April and in June. Monthly precipitation in 2010 was above and 2011 below the long-term average. See details in Table 1.

**Table 1.** Mean temperature and mean precipitation during growing seasons and long-term values.

| | Temperature (°C) | | | Precipitation (mm) | | |
|---|---|---|---|---|---|---|
| | 1980–2009 | 2010 | 2011 | 1980–2009 | 2010 | 2011 |
| March | 5.8 | 6.3 | 6.3 | 38.5 | 5.2 | 28.4 |
| April | 10.7 | 10.9 | 13.3 | 35.3 | 58.4 | 32.5 |
| May | 15.6 | 15.3 | 15.9 | 56.1 | 114.7 | 43.6 |
| June | 18.5 | 19.2 | 20.2 | 72.3 | 83.7 | 64.3 |
| July | 20.8 | 22.6 | 20.3 | 59.1 | 71.9 | 54.8 |

*2.2. Experimental Design and Management*

Pure stands of oat (cv. Effektiv) and pea (cv. Lessna) were established with 350 (oat) and 80 (pea) germinable seeds $m^{-2}$, respectively. Pea seed was not rhizobial inoculated. Three oat:pea intercrops were sown simultaneously in replacement series consisting of the following sowing ratios (oat:pea, %:%): 100:0, 75:25, 50:50, 25:75, and 0:100. For this energy efficiency study, the sowing rate was set for pure oat to 120 kg $ha^{-1}$ and for the pure pea to 210 kg $ha^{-1}$. The nitrogen fertilizer calcium ammonium nitrate (CAN, 27% N) was applied at two fertilization levels (60 and 120 kg N $ha^{-1}$) complemented by an unfertilized control. The fertilizer was applied in two equal splits, right after sowing and at the end of tillering of oat, on 2 May 2010, and on 5 May 2011. The experiment was conducted in a randomized complete block design with three replications.

Individual plots had an area of 15 $m^2$ (10 × 1.5 m) and comprised 10 rows at 12.5 cm spacing. Seedbed preparation was done with a tine cultivator to a depth of 20 cm. Sowing was performed in one pass-over with an Oyjard plot drill at a depth of 4 cm on 19 March 2010, and on 14 March 2011. The preceding crops were winter barley (2010) or spring barley (2011). Winter barley and spring barley were fertilized with 100 kg N $ha^{-1}$. Barley residues were soil incorporated. Soil mineral N at sowing was 158 (24 March 2010) and 168 (16 March 2011) kg N $ha^{-1}$ (at 0–0.9 m depth). Mechanical hand weeding was performed throughout the experiment; plants were sprayed against pests (active substance: deltamethrin).

Plants were harvested manually by cutting on the soil surface at full ripeness on 1.2 $m^2$ on 21 July 2010, and on 19 July 2011. The plant samples were divided into grain and residue. Grain and residue samples were first ground to pass through a 1 mm sieve for N determination. N concentration was measured by the Dumas combustion method using an elemental analyzer (vario MACRO cube CNS; Elementar Analysensysteme GmbH, Germany). N concentration data for grain and residues are published in Neugschwandter and Kaul [11].

*2.3. Diesel Fuel Consumption*

The diesel fuel consumption for stubble cultivation (working depth: 5–8 cm) in summer and wing sweep cultivation (working depth 16–20 cm) in autumn, seedbed preparation with power harrow (working depth 5–10 cm) and seeding with mechanical drill seeder (working depth: 4 cm) in spring was measured on a nearby field with similar soil conditions [18,19]. The wing sweep cultivator was equipped with seven tines on two bars with a tine distance of 84 cm and line distance of 42 cm and three rotary hoes for crumbling, and a wedge ring roller for crumbling and depth adjustment behind the bars of the cultivator. Working width of the wing sweep cultivator, power harrow and mechanical drill seeder was 3.0 m.

For other processes (two times mechanical weeding with spring tine harrow, spraying insecticide, harvesting with combine and transport of grain), the fuel consumptions were obtained from the Austrian Association for Agricultural Engineering and Landscape Development (ÖKL) [20]. For the transportation of the harvested grain, the diesel fuel consumption was calculated with the specific diesel fuel consumption coefficient of 0.09 L diesel fuel per ton and kilometer according to ÖKL [20] (2021). A distance of 5 km for trans-

portation of the harvested grain with a tractor and trailer was assumed. The consumption of lubrication oil consumption was set at 2% of fuel consumption [21].

Seeding and harvesting were carried out with conventional machinery (seed drill and combine harvester). The system boundary for energy analysis was defined between the field processes tillage and harvest. The additional direct and indirect energy consumption for separating of the harvested grains of the oat:pea intercrops was not considered.

### 2.4. Energy Efficiency Parameters and Energy Equivalents

The energy efficiency parameters (Table 2) were calculated according to Hülsbergen et al. [22] and Khakbazan et al. [23]. Modifications were done with special consideration of the N yield of the grain and residues. Energy in the residues was not included as an energy output since the residues were left on the field. Energy in the oat grain was set to 19.18 MJ kg$^{-1}$ dry matter and in the pea grain to 19.02 MJ kg$^{-1}$ dry matter, respectively, according to the gross energy content of feeding pea and oat grain according to the DLG-Futterwerttabelle [24]. Energy input calculations did not include seed, as the amount of seed was subtracted from the harvested grain for each crop [25]. The analysis also did not include energy associated with human labor.

**Table 2.** Definition of energy efficiency indicators.

| Parameter | Definition | Unit |
|---|---|---|
| Energy input (E) | | |
| Direct energy input ($E_d$) | $E_d$ = diesel fuel and lubricant oil | MJ ha$^{-1}$ |
| Indirect energy input ($E_i$) | $E_i$ = fertilizer + insecticide + machines | MJ ha$^{-1}$ |
| Total energy input (E) | $E = E_d + E_i$ | MJ ha$^{-1}$ |
| Energy output (EO) | | |
| EO | EO = (grain yield − seed amount) × gross energy in grain | GJ ha$^{-1}$ |
| Net-energy output (NEO) | | |
| NEO | NEO = EO − E | GJ ha$^{-1}$ |
| Energy use efficiency (EUE) | | |
| EUE | EUE = EO/E ×1000 | GJ GJ$^{-1}$ |
| Energy intensity (EI) | | |
| EI$_{\text{GRAIN-YIELD}}$ | EI$_{\text{GRAIN-YIELD}}$ = E/grain yield | MJ kg$^{-1}$ |
| EI$_{\text{AGB-YIELD}}$ | EI$_{\text{AGB-YIELD}}$ = E/AGB $^A$ | MJ kg$^{-1}$ |
| EI$_{\text{GRAIN\_N-YIELD}}$ | EI$_{\text{GRAIN\_N-YIELD}}$ = E/grain N yield | MJ kg$^{-1}$ |
| EI$_{\text{AGB\_N-YIELD}}$ | EI$_{\text{AGB\_N-YIELD}}$ = E/AGB N yield | MJ kg$^{-1}$ |
| Energy productivity (EP) | | |
| EP$_{\text{GRAIN-YIELD}}$ | EP$_{\text{GRAIN-YIELD}}$ = grain yield/E | kg MJ$^{-1}$ |
| EP$_{\text{RES-YIELD}}$ | EP$_{\text{RES-YIELD}}$ = residues yield/E | kg MJ$^{-1}$ |
| EP$_{\text{AGB-YIELD}}$ | EP$_{\text{AGB-YIELD}}$ = AGB yield/E | kg MJ$^{-1}$ |
| EP$_{\text{GRAIN\_N-YIELD}}$ | EP$_{\text{GRAIN\_N-YIELD}}$ = grain N yield/E | g N MJ$^{-1}$ |
| EP$_{\text{RES\_N-YIELD}}$ | EP$_{\text{RES\_N-YIELD}}$ = residues N yield/E | g N MJ$^{-1}$ |
| EP$_{\text{AGB\_N-YIELD}}$ | EP$_{\text{AGB\_N-YIELD}}$ = AGB N yield/E | g N MJ$^{-1}$ |

$^A$ Above-ground biomass, RES = residues, Crop yield in dry matter.

The determination of the energy equivalent of the indirect energy in farm machinery was done based on a 100 ha cereal area under Austrian conditions by Biedermann [26]. For this calculation, different estimated technical and economic lifetimes of the machinery were assumed: 10,000 h for the tractor, 3000 h for the combine harvester, and 2000–3000 ha for the implements.

The amounts of the used production facilities were multiplied by the energy equivalents (Table 3).

**Table 3.** Energy equivalents for production facilities.

|  | Unit | Energy Equivalent | References |
|---|---|---|---|
| **Direct energy use** | | | |
| Diesel fuel | MJ L$^{-1}$ | 39.6 | [22,27] |
| Lubricant oil | MJ L$^{-1}$ | 39.0 | [27] |
| **Indirect energy use** | | | |
| Mineral fertilizer: Calcium ammonium nitrate (27% N) | MJ kg$^{-1}$ N | 32.2 | [28,29] |
| Insecticide: Deltamethrin | MJ kg$^{-1}$ a.i. [B] | 217 | [25] |
| Machinery: Conservation tillage [A] | MJ ha$^{-1}$ | 1810 | [26] |

[A] Energy equivalent of the weight for tractor: 65 MJ kg$^{-1}$, tillage implement: 48 MJ kg$^{-1}$, spreader and sprayer: 55 MJ kg$^{-1}$, combine harvester: 70 MJ kg$^{-1}$; [B] Active ingredient.

*2.5. Statistical Analysis*

All analyses were conducted using the IBM® SPSS® Statistics 21. The requirements for analysis of variance (ANOVA) were tested with the Levene test for homogeneity of variances and Shapiro-Wilk test for normal distribution of residuals. ANOVA tests were carried out for crop yield, N yield, energy output, net-energy output, energy intensity, energy productivity and energy use efficiency to detect year, sowing ratio and N fertilization effects. Multiple comparisons to separate means were carried out with the Student-Newman-Keuls procedure ($p < 0.05$).

**3. Results**

*3.1. Total Fuel Consumption and Energy Input*

The total area-based diesel fuel consumption for the field processes in the unfertilized control was 64.1 L ha$^{-1}$ and is made up of 5.7 L ha$^{-1}$ for stubble cultivation, 9.4 L ha$^{-1}$ for ground cultivation, 8.6 L ha$^{-1}$ for seedbed preparation, 6.3 L ha$^{-1}$ for seeding, 7.0 L ha$^{-1}$ for mechanical weeding, 2.0 L ha$^{-1}$ for spraying insecticide, 22.9 L ha$^{-1}$ for harvesting with combine, and 2.2 L ha$^{-1}$ for grain transport. N-fertilization (60 kg N ha$^{-1}$ and 120 kg N ha$^{-1}$) with two equal splits required an additional 3.0 L ha$^{-1}$ in total.

Due to missing N-fertilization in the control variant, the fuel consumption and direct energy input were lower than in the 60 and 120 kg N ha$^{-1}$ level. The ratio of direct energy to indirect energy (in %:%) was: 59:41 for 0 kg N ha$^{-1}$, 42:58 for 60 kg N ha$^{-1}$ and 32:68 for 120 N ha$^{-1}$ (Table 4). The indirect and total energy input increased with increasing N fertilization levels. The N fertilizer energy reached almost 50% of the total energy input in the fertilization level of 120 kg N ha$^{-1}$.

**Table 4.** Direct and indirect energy input (MJ ha$^{-1}$) for oat:peat intercrops as affected by N fertilization.

| N Fertilization | Direct Energy | Indirect Energy | | | Sum |
|---|---|---|---|---|---|
|  |  | N Fertilizer | Insecticide | Machinery |  |
| 0 | 2588 | 0 | 2 | 1810 | 4400 |
| 60 | 2709 | 1932 | 2 | 1810 | 6453 |
| 120 | 2709 | 3864 | 2 | 1810 | 8385 |

*3.2. Crop Yields and N Yields*

Mean values of yields and N yields of crops stands as affected by main factor effects sowing ratio, N fertilizer level and year are shown in Table 5 and detected factor interactions are shown in Tables 6 and 7.

**Table 5.** Crop yields and N yields of oat:pea intercrops as affected by sowing ratio, N fertilizer level and year.

| | Crop Yields | | | Nitrogen Yield | | |
|---|---|---|---|---|---|---|
| | **Grain** | **Residues** | **AGB** [A] | **Grain** | **Residues** | **AGB** [A] |
| | (kg ha$^{-1}$) | | | | | |
| Sowing ratio (oat:pea, %:%) | | | | | | |
| 100:0 | 5081 [b] | 6961 [b] | 12,042 | 109.5 [ab] | 44.3 [a] | 153.8 [a] |
| 75:25 | 4660 [bc] | 7403 [b] | 12,063 | 107.8 [a] | 53.9 [b] | 161.7 [a] |
| 50:50 | 4543 [ab] | 6963 [b] | 11,506 | 112.3 [ab] | 57.9 [bc] | 169.3 [a] |
| 25:75 | 4425 [a] | 6621 [b] | 11,046 | 124.2 [b] | 66.6 [c] | 190.8 [b] |
| 0:100 | 5569 [c] | 5265 [a] | 10,834 | 205.4 [c] | 65.9 [c] | 271.3 [c] |
| Fertilization (kg N ha$^{-1}$) | | | | | | |
| 0 | 4489 [a] | 5964 [a] | 10,453 [a] | 118.2 [a] | 42.8 [a] | 161.0 [a] |
| 60 | 5085 [b] | 6927 [b] | 12,012 [b] | 134.9 [b] | 58.1 [b] | 193.0 [b] |
| 120 | 4993 [b] | 7037 [b] | 12,030 [b] | 142.5 [b] | 72.7 [c] | 215.2 [c] |
| Year | | | | | | |
| 2010 | 4836 | 6974 [b] | 11,810 | 130.7 | 69.9 [b] | 200.6 [b] |
| 2011 | 4875 | 6311 [a] | 11,187 | 133.0 | 46.6 [a] | 179.6 [a] |
| ANOVA | | | | | | |
| Sowing ratio (SR) | *** | *** | | *** | *** | *** |
| Fertilization (F) | ** | *** | *** | *** | *** | *** |
| Year (Y) | | ** | | | *** | ** |
| SR × F | | | | | ** | |
| SR × Y | | * | | ** | * | ** |
| F × Y | | | | | * | |

[A] Above-ground biomass; Crop yields in dry matter; Significance level: $p < 0.05$ (*), $p < 0.01$ (**), $p < 0.001$ (***). There were no statistically significant SR × F × Y interactions. The small letters in the tables show the significant differences.

**Table 6.** Residues N yield of oat and pea pure crop stands and oat:pea intercrops as affected by sowing ratio and N fertilizer.

| Fertilization (kg N ha$^{-1}$) | Sowing Ratio (oat:pea; %:%) | | | | | LSD [A] |
|---|---|---|---|---|---|---|
| | **100:0** | **75:25** | **50:50** | **25:75** | **0:100** | |
| Residues N yield (kg ha$^{-1}$) | | | | | | |
| 0 | 28 | 35 | 45 | 52 | 54 | |
| 60 | 45 | 43 | 62 | 65 | 73 | 15 |
| 120 | 60 | 87 | 68 | 80 | 71 | |

[A] Least significant difference.

Mean values over all sowing ratios, N fertilization levels and years were as follows: grain yield: 4856 kg ha$^{-1}$, residue yield: 6643 kg ha$^{-1}$, above ground biomass (AGB) yield: 11,499 kg ha$^{-1}$, grain N yield: 131.9 kg ha$^{-1}$, residue N yield: 57.7 kg ha$^{-1}$ and AGB N yield: 190.0 kg ha$^{-1}$.

The grain yield was highest in pure pea and also high in the intercrops with the sowing ratio of 75:25 and lowest in the intercrops with the 25:75 sowing ratio, pure oat showed an intermediate value (Table 5). Both fertilization treatments increased the grain yield, with no differences between 60 and 120 kg N ha$^{-1}$. No influence of the year was observed on the grain yield. The residue yield was in pure oat and in all oat:pea intercrops higher than in pure pea. There was a statistically significant fertilization × year interaction: The residue yield was in 2010 lowest in pure pea, whereas it decreased in 2011 with a lower pea share on the sowing ratio. The AGB was higher with 60 and 120 kg N ha$^{-1}$ compared to the control. Both the sowing ratio and the year had no influence on the AGB (Table 5).

**Table 7.** Residues yield and N yields of oat and pea pure crop stands and oat:pea intercrops as affected by sowing ratio and year.

| Year | Sowing Ratio (oat:pea, %:%) | | | | | LSD [A] |
|---|---|---|---|---|---|---|
| | 100:0 | 75:25 | 50:50 | 25:75 | 0:100 | |
| Residues yield (kg ha$^{-1}$) | | | | | | |
| 2010 | 6957 | 7278 | 7332 | 7315 | 5990 | 922 |
| 2011 | 6966 | 7527 | 6594 | 5928 | 4541 | |
| Grain N yield (kg ha$^{-1}$) | | | | | | |
| 2010 | 108 | 101 | 104 | 119 | 222 | 21 |
| 2011 | 111 | 115 | 121 | 129 | 189 | |
| Residues N yield (kg ha$^{-1}$) | | | | | | |
| 2010 | 47 | 64 | 70 | 85 | 84 | 12 |
| 2011 | 42 | 46 | 47 | 50 | 48 | |
| AGB N yield [B] (kg ha$^{-1}$) | | | | | | |
| 2010 | 155 | 166 | 174 | 203 | 306 | 28 |
| 2011 | 153 | 161 | 168 | 179 | 237 | |

[A] Least significant difference; [B] Above-ground biomass; Residues yield in dry matter.

The grain N yield was statistically significant affected by the sowing ratio × year (Table 5): It was highest in pure pea (with a higher value in 2010 than in 2011) and did not differ between pure oat and intercrops (Table 7). Both fertilization treatments increased the grain N yield, with no differences between 60 and 120 kg N ha$^{-1}$ (Table 5).

The residues N yield was statistically significant affected by interactions of sowing ratio × fertilization, sowing ratio × year, and fertilization × year (Table 5): The residues N yield increased with a higher share of pea on the sowing ratios, especially in the control; the increase with N fertilization was higher with a higher oat share (Table 6). It was, except for pure oat, higher in 2010 than in 2011 (Table 7). The increase in fertilization was with values for 0, 60, and 120 kg N ha$^{-1}$ in 2010 of 51, 71 and 90 kg N ha$^{-1}$ and in 2011 of 36, 46 and 58 kg N ha$^{-1}$ (LSD = 10), higher in 2010 than in 2011 (Table 7). The AGB N yield was statistically significant affected by the sowing ratio × year (Table 5): It was highest in pure pea in both years, with higher values in pure pea than in all other sowing ratios in 2011, whereas in 2011, the 25:75 intercrops showed intermediate values between pure pea and the other sowing ratios (Table 7). Both fertilization treatments increased the AGB N yield, with a higher increase of 120 compared to 60 kg N ha$^{-1}$ (Table 5).

### 3.3. Energy Efficiency for Biomass Yield and N Yield

The mean values of energy efficiency for biomass yields and N yields as affected by the main factor effects sowing ratio, N fertilizer level and year, are shown in Table 8. Detected factor interactions are shown in Tables 9–11.

Mean values over all sowing ratios, N fertilization levels and years were as follows: EO: 90.1 GJ ha$^{-1}$, NEO: 84.0 GJ ha$^{-1}$, EUE: 14.9 GJ GJ$^1$, EI$_{GRAIN-YIELD}$: 1.33 MJ kg$^{-1}$, EI$_{AGB-YIELD}$: 0.56 MJ kg$^{-1}$, EI$_{GRAIN\_N-YIELD}$: 52.6 MJ kg$^{-1}$, EI$_{AGB\_N-YIELD}$: 35.3 MJ kg$^{-1}$, EP$_{GRAIN-YIELD}$: 0.80 kg MJ$^{-1}$, EP$_{RES-YIELD}$: 1.09 kg MJ$^{-1}$, EP$_{AGB-YIELD}$: 1.89 kg MJ$^{-1}$, EP$_{GRAIN\_N-YIELD}$: 21.59 g N MJ$^{-1}$, EP$_{RES\_N-YIELD}$: 9.22 g N MJ$^{-1}$ and EP$_{AGB\_N-YIELD}$: 30.8 g N MJ$^{-1}$.

The EO, NEO, and EUE were in oat:pea intercrops lower than in pure stands of oat and pea (except for the 75:25 intercrops, which did not significantly differ from pure pea). N fertilization increased the EO, NEO, and EI. The EO did not differ between both fertilization levels. The NEO was highest with 60 kg N ha$^{-1}$ and lowest in the control with 120 kg N ha$^{-1}$ showing intermediate values. The EUE increased from the control over 60 to 120 kg N ha$^{-1}$. The year did not affect these energy efficiency parameters (Table 8).

**Table 8.** Energy efficiency parameters of oat and pea pure crop stands and oat:pea intercrops as affected by sowing ratio, N fertilizer level and year.

| | EO [A] | NEO [B] | EUE [C] | EI [D] | | | | EP [E] | | | | | |
|---|---|---|---|---|---|---|---|---|---|---|---|---|---|
| | | | | EI$_{GRAIN-YIELD}$ | EI$_{AGB-YIELD}$ | EI$_{GRAIN\_N-YIELD}$ | EI$_{AGB\_N-YIELD}$ | EP$_{GRAIN-YIELD}$ | EP$_{RES-YIELD}$ | EP$_{AGB-YIELD}$ | EP$_{GRAIN\_N-YIELD}$ | EP$_{RES\_N-YIELD}$ | EP$_{AGB\_N-YIELD}$ |
| | (MJ ha$^{1}$) | | GJ GJ$^{-1}$ | (MJ kg$^{-1}$) | | | | (kg MJ$^{-1}$) | | | (g N MJ$^{-1}$) | | |
| **Sowing ratio (oat:pea, %:%)** | | | | | | | | | | | | | |
| 100:0 | 95.4 [bc] | 89.0 [bc] | 15.4 [bc] | 1.27 [ab] | 0.53 [a] | 59.1 [bc] | 41.9 [d] | 0.82 [a] | 1.14 [b] | 1.96 | 17.33 [a] | 6.84 [a] | 24.17 [a] |
| 75:25 | 86.9 [ab] | 80.5 [ab] | 14.5 [ab] | 1.44 [b] | 0.54 [a] | 61.1 [c] | 39.2 [cd] | 0.78 [a] | 1.19 [b] | 2.00 | 17.63 [a] | 8.39 [b] | 26.02 [ab] |
| 50:50 | 84.3 [a] | 77.8 [a] | 14.1 [ab] | 1.44 [b] | 0.56 [ab] | 58.8 [bc] | 37.4 [c] | 0.76 [a] | 1.15 [b] | 1.91 | 18.88 [a] | 9.43 [bc] | 28.31 [bc] |
| 25:75 | 81.5 [a] | 75.0 [a] | 13.4 [a] | 1.46 [b] | 0.59 [ab] | 52.2 [b] | 33.7 [b] | 0.73 [a] | 1.10 [b] | 1.82 | 20.37 [a] | 10.75 [c] | 31.13 [c] |
| 0:100 | 102.4 [d] | 96.0 [d] | 16.9 [c] | 1.17 [a] | 0.60 [b] | 32.0 [a] | 24.5 [a] | 0.92 [b] | 0.86 [a] | 1.78 | 33.75 [b] | 10.68 [c] | 44.43 [d] |
| **Fertilization (kg N ha$^{-1}$)** | | | | | | | | | | | | | |
| 0 | 83.1 [a] | 78.7 [a] | 18.9 [c] | 1.00 [a] | 0.43 [a] | 41.1 [a] | 29.6 [a] | 1.02 [c] | 1.36 [c] | 2.38 [c] | 26.86 [c] | 9.96 [b] | 36.82 [c] |
| 60 | 94.5 [b] | 88.0 [b] | 14.6 [b] | 1.31 [b] | 0.55 [b] | 52.2 [b] | 36.0 [b] | 0.79 [b] | 1.07 [b] | 1.86 [b] | 20.90 [b] | 8.89 [a] | 29.79 [b] |
| 120 | 92.7 [b] | 84.4 [ab] | 11.1 [a] | 1.76 [c] | 0.71 [c] | 64.6 [c] | 40.4 [c] | 0.60 [a] | 0.84 [a] | 1.43 [a] | 17.01 [a] | 8.81 [a] | 25.82 [a] |
| **Year** | | | | | | | | | | | | | |
| 2010 | 89.7 | 83.3 | 14.9 | 1.38 | 0.55 [a] | 54.8 [b] | 33.9 [a] | 0.80 | 1.14 [b] | 1.94 | 21.62 | 11.09 [b] | 32.71 [b] |
| 2011 | 90.5 | 84.0 | 14.8 | 1.33 | 0.58 [b] | 50.5 [a] | 36.8 [b] | 0.80 | 1.04 [a] | 1.84 | 21.57 | 7.35 [a] | 28.92 [a] |
| **ANOVA** | | | | | | | | | | | | | |
| Sowing ratio (SR) | *** | *** | *** | *** | | *** | *** | *** | *** | | *** | *** | *** |
| Fertilization (F) | ** | * | *** | *** | *** | *** | *** | *** | *** | *** | *** | * | *** |
| Year (Y) | | | | | | * | * | ** | | ** | | *** | *** |
| SR × F | | | | * | | ** | | | | | ** | *** | ** |
| SR × Y | | | | | | | | | | | ** | ** | ** |
| F × Y | | | | * | | * | | | | | | | |

[A] Energy output, [B] Net-energy output, [C] Energy use efficiency, [D] Energy intensity—crop yield in dry matter, [E] Energy productivity—crop yield in dry matter, AGB = Above-ground biomass, RES = Residues; Significance level: $p < 0.05$ (*), $p < 0.01$ (**), $p < 0.001$ (***). There were no statistically significant SR × F × Y interactions. The small letters in the tables show the significant differences.

**Table 9.** Energy efficiency parameters of oat and pea pure crop stands and oat:pea intercrops as affected by sowing ratio and N fertilizer.

| Fertilization (kg N ha$^{-1}$) | | Sowing Ratio (oat:pea, %:%) | | | | | LSD [C] |
|---|---|---|---|---|---|---|---|
| | | 100:0 | 75:25 | 50:50 | 25:75 | 0:100 | |
| EI$_{GRAIN-YIELD}$ [A] | (MJ kg$^{-1}$) | | | | | | |
| 0 | | 1.04 | 1.02 | 0.97 | 1.12 | 0.87 | |
| 60 | | 1.27 | 1.23 | 1.50 | 1.38 | 1.14 | 0.26 |
| 120 | | 1.48 | 2.08 | 1.85 | 1.89 | 1.49 | |
| EI$_{GRAIN\_N-YIELD}$ [A] | (MJ kg$^{-1}$) | | | | | | |
| 0 | | 55.4 | 47.6 | 38.4 | 39.7 | 24.2 | |
| 60 | | 58.9 | 54.3 | 63.3 | 51.9 | 32.8 | 11.1 |
| 120 | | 63.2 | 81.4 | 74.6 | 65.0 | 39.1 | |
| EP$_{GRAIN\_N-YIELD}$ [B] | (g N MJ$^{-1}$) | | | | | | |
| 0 | | 18.5 | 21.1 | 26.4 | 25.6 | 42.5 | |
| 60 | | 17.3 | 18.5 | 16.5 | 19.6 | 32.6 | 4.2 |
| 120 | | 16.1 | 13.2 | 13.7 | 15.9 | 26.1 | |
| EP$_{RES\_N-YIELD}$ [B] | (g N MJ$^{-1}$) | | | | | | |
| 0 | | 6.5 | 8.0 | 10.3 | 12.6 | 12.4 | |
| 60 | | 7.0 | 6.5 | 9.6 | 10.1 | 11.3 | 2.1 |
| 120 | | 7.1 | 10.6 | 8.3 | 9.5 | 8.4 | |
| EP$_{AGB\_N-YIELD}$ [B] | (g N MJ$^{-1}$) | | | | | | |
| 0 | | 25.0 | 29.2 | 36.8 | 38.3 | 54.9 | |
| 60 | | 24.2 | 25.0 | 26.1 | 29.7 | 43.9 | 5.3 |
| 120 | | 23.3 | 23.9 | 22.0 | 25.5 | 34.5 | |

[A] Energy intensity, [B] Energy productivity, [C] Least significant difference, AGB = Above-ground biomass, RES = Residues.

**Table 10.** Energy efficiency parameters of oat and pea pure crop stands and oat:pea intercrops as affected by sowing ratio and year.

| Year | | Sowing Ratio (oat:pea, %:%) | | | | | LSD [B] |
|---|---|---|---|---|---|---|---|
| | | 100:0 | 75:25 | 50:50 | 25:75 | 0:100 | |
| EP$_{GRAIN\_N-YIELD}$ [A] | (g N MJ$^{-1}$) | | | | | | |
| 2010 | | 17.16 | 16.76 | 17.52 | 19.85 | 36.78 | |
| 2011 | | 17.49 | 18.50 | 20.24 | 20.90 | 30.71 | 3.46 |
| EP$_{RES\_N-YIELD}$ [A] | (g N MJ$^{-1}$) | | | | | | |
| 2010 | | 7.25 | 9.87 | 11.32 | 13.54 | 13.46 | |
| 2011 | | 6.44 | 6.91 | 7.53 | 7.97 | 7.92 | 1.72 |
| EP$_{AGB\_N-YIELD}$ [A] | (g N MJ$^{-1}$) | | | | | | |
| 2010 | | 24.41 | 26.63 | 28.85 | 33.39 | 50.25 | |
| 2011 | | 23.92 | 25.41 | 27.77 | 28.87 | 38.62 | 4.34 |

[A] Energy productivity, [B] Least significant difference, AGB = Above-ground biomass, RES = Residues.

The EI$_{GRAIN-YIELD}$ did not differ between sowing ratios in the unfertilized variants, it was higher in the 50:50 intercrop than in the 75:25 intercrop and pure pea with 60 kg N ha$^{-1}$ and in all intercrops than in the pure crop stand with 120 kg ha$^{-1}$ (Table 9). The EI$_{GRAIN-YIELD}$ increased with fertilization, with a stronger increase in 2010 than in 2011 (Table 11). The EI$_{AGB-YIELD}$ did not differ between the sowing ratios. It was ranked among fertilization levels as follows: 0 < 60 < 120 kg N ha$^{-1}$.

**Table 11.** Energy efficiency parameters of oat and pea pure crop stands and oat:pea intercrops as affected by N fertilizer level and year.

| Year | | Fertilization (kg N ha$^{-1}$) | | | LSD [B] |
|---|---|---|---|---|---|
| | | 0 | 60 | 120 | |
| EI$_{GRAIN-YIELD}$ [A] | (MJ kg$^{-1}$) | | | | |
| 2010 | | 0.98 | 1.30 | 1.87 | 0.17 |
| 2011 | | 1.03 | 1.31 | 1.64 | |
| EI$_{GRAIN\_N-YIELD}$ [A] | (MJ kg$^{-1}$) | | | | |
| 2010 | | 40.3 | 53.8 | 70.2 | 7.1 |
| 2011 | | 41.8 | 50.7 | 59.1 | |

[A] Energy intensity, [B] least significant difference, grain yield in dry matter.

The EI$_{GRAIN\_N-YIELD}$ was lowest for pure pea and increased with an increasing oat share in the intercrops (Table 8). Highest values very found in pure oat and in 75:25 intercrops (Table 8). The EI$_{GRAIN\_N-YIELD}$ increased with fertilization, with a stronger increase in 2010 than in 2011 (Table 11). The EI$_{AGB-N-YIELD}$ was highest in pure oat, decreased with higher pea and lower oat share in the intercrops and was lowest in pure pea. It was ranked among fertilization levels as followed: $0 < 60 < 120$ kg N ha$^{-1}$ (Tables 8 and 9) EI$_{GRAIN\_N-YIELD}$ and EI$_{AGB-N-YIELD}$ were higher in 2010 than in 2011 (Table 8).

EI$_{GRAIN-YIELD}$ and EI$_{GRAIN\_N-YIELD}$ showed significant interactions of year$\times$fertilization, where the separated means are shown in Table 9. Whereas the fertilization factor was significant in each year, in the dry year 2011, the EI$_{GRAIN-YIELD}$ and EI$_{GRAIN\_N-YIELD}$ were significantly lower with the fertilization rate 120 kg N ha$^{-1}$.

Pure stands of pea had the highest EP$_{GRAIN-YIELD}$ and the lowest EP$_{RES-YIELD}$ compared to pure oat and all intercrops (Table 8). EP$_{AGB-YIELD}$ was not affected by the sowing ratio. The EP$_{GRAIN-YIELD}$ and the EP$_{RES-YIELD}$ decreased with N fertilization and were ranked among fertilization levels as followed: $0 > 60 > 120$ kg N ha$^{-1}$. The EP$_{AGB-YIELD}$ was lowest with 120 kg N ha$^{-1}$. EP$_{GRAIN-YIELD}$ and EP$_{AGB-YIELD}$ did not differ between years, whereas EP$_{RES-YIELD}$ was higher in 2010 than in 2011 (Table 8).

EP$_{GRAIN\_N-YIELD}$, EP$_{RES\_N-YIELD}$ and EP$_{ABG\_N-YIELD}$ were generally highest in pure pea, decreased with lower pea and higher oat share in the intercrops and had lowest values in pure oat (Table 8). All parameters were higher with pea shares on the sowing ratios and lower with increasing N fertilization. The differences between the fertilization treatments decreased with increasing oat share on the sowing ratios, resulting in no differences of EP$_{GRAIN\_N-YIELD}$, EP$_{RES\_N-YIELD}$ and EP$_{ABG\_N-YIELD}$ between the N treatment in pure oat (Table 9). EP$_{GRAIN\_N-YIELD}$, EP$_{RES\_N-YIELD}$ and EP$_{ABG\_N-YIELD}$ of pure oat did not differ between years (Table 8). With an increasing share of pea on the sowing ratios, the values of all three parameters increased, with a higher increase in 2010 than in 2011 (Table 9). The EP$_{GRAIN\_N-YIELD}$ decreased in both years with a higher N fertilization, with a higher decrease in 2010 than in 2011. EP$_{GRAIN\_N-YIELD}$ did not differ between years, whereas EP$_{RES-YIELD}$ and EP$_{ABG\_N-YIELD}$ were higher in 2010 than in 2011 (Table 8).

With increasing pea share, the EP$_{GRAIN\_N-YIELD}$, EP$_{RES\_N-YIELD}$ and EP$_{ABG\_N-YIELD}$ for N increased (Table 8).

The significant interactions of year $\times$ sowing rate for EP$_{GRAIN\_N-YIELD}$, EP$_{RES\_N-YIELD}$ and EP$_{AGB\_N-YIELD}$ are presented in Table 10. EP$_{GRAIN\_N-YIELD}$ was in pure pea stands (0:100) in the dry year 2011, significantly lower than in 2010. EP$_{RES\_N-YIELD}$ was lower in 2011 than in 2010 for the sowing rations (not significant for pure oat stands). Only the sowing ratios 100:0 and 75:25 in the year 2010 were significant different for EP$_{RES\_N-YIELD}$. EP$_{AGB\_N-YIELD}$ was lower in the dry year 2011 than in 2010 (significant for the sowing ratio 25:75 and 0:100). Pure stands of pea showed in all years significantly higher EP$_{AGB\_N-YIELD}$ than for the other sowing rates.

## 4. Discussion

### 4.1. Total Fuel Consumption and Energy Input

Technical energy input as a direct source (fuel) and indirect source (fertilizer, pesticides, machinery) is a crucial indicator for the intensity of plant production.

The total area-based diesel fuel consumption for the tillage processes was similar to conservation-tilled cereals in the Pannonian region [30]. The mechanical weeding of the intercrops requires more diesel fuel energy than chemical weeding with herbicides. Additional mineral N fertilization with a spreader required only a small diesel energy amount for application in comparison to fertilization with organic manure (farm yard manure, compost, slurry) [31]. The total area-based energy input was mainly determined by the amount of mineral N fertilization. The sowing ratio did not influence the fuel consumption and energy input because the intercrops and pure crop stands were both sown with a mechanical seed drill in one process. Whereas the amount the N fertilization affected the indirect and total energy input significantly. It is well known that mineral N fertilization is a significant management factor, which is highly contributing to the energy input in cropping systems [9,32,33]. The total area-based energy input is a well known indicator of the intensity of crop production [22,34]. In our study, the energy inputs were much lower than the threshold value for a low-input arable farming system with 10 GJ ha$^{-1}$ [34].

### 4.2. Crop Yields and N Yields

The pure pea crop stands had the highest grain yields but the lowest residue yields. However, both the grain N yields, and the residue N yields were highest in pure pea and in the intercrops with the highest pea share. All these parameters increased with N fertilization.

### 4.3. Energy Efficiency for Biomass Yield and N Yield

The net-energy output (=energy gain) is, according to Arvidsson [29], the most relevant parameter in determining the efficiency of cropping systems in a world of increasing food and energy demand. In this consideration, pure stands of oat are more energy efficient than oat:pea intercrops and pure stands of pea. Hülsbergen et al. [22] used the parameters energy use efficiency and energy intensity for determining optimum input levels from an ecological point of view. In our study, pure stands of oat showed the highest EUE and lowest EI$_{\text{GRAIN-YIELD}}$. Similar to the area-based energy consumption, also the product based energy consumption (=energy intensity) was increased with mineral N fertilization.

Pure stands had a higher EUE and EI$_{\text{GRAIN-YIELD}}$ than intercrops, indicating that pure stands used the growing factors (nutrient, water, photosynthetic radiation) more efficiently than intercrops. The range between the lowest and highest EI$_{\text{GRAIN-YIELD}}$ was highest with 120 kg N ha$^{-1}$ and lowest in the control, indicating that N fertilization had a stronger response than sowing ration. The EUE was higher in pure pea than in pure oat. A higher EUE of grain legumes than cereal grains was also found in the energy efficiency analyses of crops in a long-term tillage experiment at the location [30,35]. The range of EUE was two times higher in fertilization than in the sowing ratio, indicating that the N fertilization mainly determines EUE than by the sowing ratio. A higher EUE can be achieved with low N fertilization.

The indicator EI for N yield allows the energetic evaluation of the N yield. One goal in plant production is also to harvest N in grain, residues and AGB. Our study showed that the N uptake in the AGB (=EI$_{\text{AGB\_N-YIELD}}$) was in pure oat stands more energy intensive than oat:pea intercrops and pure pea stands. With increasing pea share in the oat:pea intercrops, the EI$_{\text{GRAIN\_N-YIELD}}$, EI$_{\text{STRAW\_N-YIELD}}$ and the EI$_{\text{AGB-N-YIELD}}$ decreased, which means that the N uptake by the AGB (N yield) requires in legumes and also in the intercrops with a legume less technical energy than the cereals. The reason for that is the contribution of N through biological N fixation of the legume in both the intercrops and the pure crop stand [15].

The energy productivity for AGB was much more affected by the factor N fertilization than by the factor sowing ratio, showing that $EP_{AGB-YIELD}$ responded stronger to N fertilization than to the sowing ratio. With 1 MJ technical energy (direct and indirect energy input), the pure pea crop stands could produce the highest grain yield and lowest residue yield. In comparison, the amount of AGB produced with 1 MJ of technical energy was the same for pure crop stands and intercrops. The energy productivity for the N yield was not clearly explained by the main factors (N fertilization and sowing rate) because of significant interactions. Generally, a higher N yield resulted in higher energy productivity for N yield.

The overall energy productivity of the N yield of the AGB ($EP_{AGB\_N-Yield}$) in the cropping systems was 30.8 g N $MJ^{-1}$, similar to the technical energy productivity in mineral N fertilizer (calcium ammonium nitrate: 31.1 g N $MJ^{-1}$ = reciprocal value of 32.2 MJ $kg^{-1}$ N, Table 3). Still, there is a large range of the $EP_{AGB\_N-Yield}$ from 22.0 to 54.9 g N $MJ^{-1}$ (Table 9). A more realistic approach would be to compare the $EP_{RES\_N-YIELD}$ for zero N fertilization with the technical energy productivity in mineral N fertilizer. According to our study, about 10 g N could be produced in the plant residues with 1 MJ technical energy input, which is about 68% lower than the technical energy productivity in mineral N fertilizer, without consideration of the technical energy input for spreading. With increasing N fertilization, the range of the lowest to the highest $EP_{RES\_N-YIELD}$ decreased, indicating that $EP_{RES\_N-YIELD}$ responded more by N fertilization than the sowing ratio.

Additionally, the weather conditions during the vegetation period significantly determine the indicators of energy productivity ($EP_{GRAIN\_N-YIELD}$, $EP_{RES\_N-YIELD}$, $EP_{AGB-YIELD}$). Weather conditions with enough precipitation during the vegetation period (the year 2010) resulted in higher energy productivity than in the dry year 2011. Depending on the soil water availability, the soil mineral N from different sources (organic and mineral fertilizer, biological nitrogen fixation, humus, N deposition) is mainly responsible for the biomass production and energy output. It is supposed that the energy efficiency indicators will be affected positively, if the soil mineral N is not delivered by the energy-intensive mineral N fertilizer.

The benefits of intercropping systems are well-known [7], but our study showed that these systems are not system immanent more energy efficient. The energy input could also be higher if the further technical energy input for the separating of the harvested intercrops seeds had been considered. This would impair the energy efficiency indicators.

## 5. Conclusions

In the Pannonian region, where the soil water content mainly determines plant growth, the energy efficiency of cropping systems plays an important role in arable farming. Oat:pea intercrops increase the in-field biodiversity and have many well-known ecological benefits. However, there exists a trade-off between the benefits of the in-field biodiversity and energy efficiency. High energy efficiency could be better reached with pure stands of legumes than with oat:pea intercrops. The sowing ratio is a management tool for farmers to optimize between in-field biodiversity and energy efficiency. The N in plant residues of legumes is produced with lower energy than from cereal straw. With a higher share of legumes in the intercrop, the energy productivity of N in the plant residues increased. From the point of energy efficiency for the biologically fixed N, it is better to have a higher degree of legumes in the intercrops. The N yield in the legume residues, which is available for subsequent crops, requires 68% lower technical energy than the N in mineral fertilizer. From this point of view, legumes in intercrops are energy efficient N producers within crop rotations.

**Author Contributions:** Conceptualization, G.M. and R.W.N.; methodology, G.M. and R.W.N.; software, G.M.; validation, G.M. and J.B.; formal analysis, G.M.; investigation, G.M.; resources, H.W.; data curation, R.W.N.; writing—original draft preparation, G.M. and R.W.N.; writing—review and editing, G.M., H.-P.K. and J.B.; visualization, G.M.; supervision, H.-P.K. and H.W.; project administration, G.M. and H.W. All authors have read and agreed to the published version of the manuscript.

**Funding:** This work was supported by the Austrian Federal Ministry of Education, Science and Research (Project No. CZ 05/2022) and Czech Ministry of Education, Youth and Sports within the Scientific & Technological Cooperation (Project No. 8J22AT001).

**Institutional Review Board Statement:** Not applicable.

**Informed Consent Statement:** Not applicable.

**Data Availability Statement:** Not applicable.

**Acknowledgments:** The authors thank the technical staff of the experimental farm Groß-Enzersdorf of the University of Life Sciences and Natural Resources (BOKU) for conducting the field experiments.

**Conflicts of Interest:** The authors declare no conflict of interest.

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
