# Peer review of "Energy Efficiency of Oat:Pea Intercrops Affected by Sowing Ratio and Nitrogen Fertilization"

_agronomy, doi:10.3390/agronomy13010042_

Round 1

Reviewer 1 Report

The reviewed manuscript presents interesting research. However, I have some very serious remarks and comments. Therefore, at this stage, I propose to reject the manuscript and encourage authors to resubmit it after its verification.

L84 - It is puzzling why these results are being published only now, since the research was carried out in 2010-2011, i.e. over 12 years ago? It needs to be clarified whether anything has changed in this regard over such a long period of time. Have these crops entered commercial production? In addition, is oats not a demanding plant for such good soil? Shouldn't a more demanding crop, such as wheat, be grown here?

L141 – It is necessary to add information and formulas on how the biomass yield, N content and N yield of the grain and residues were calculated. Therefore, after the chapter "2.4. Energy efficiency parameters and energy equivalents", a chapter on the methodology for determining biomass yields, N content and N yields should be added.

L147-148. What does it mean that "it was considered as a negligible source of 148 energy" please specify what range or percentage it may be in relation to, for example, other studies that mention it.

The entire section “3.2. Crop yields and N yields”. It is necessary to clarify and specify whether crop yields (grain, reidues, AGB) are given in absolutely dry matter (dry matter DM) or at the moisture content that was present during harvesting, i.e. fresh matter (Fresh Matter FM), and then the moisture content of the harvested biomass should be given. It is also necessary to check the calculations and roundings in the whole mansuscript, because e.g. in Table 5, variant 100:0 (GY 5081+RY 6963 = AGB 12044, not 12043 kg/ha). The same is true for N yield (GNY 109.5+RNY 44.9 = AGB 154.4, not 153.8 kg/ha). Therefore, it is necessary to check all calculations in the entire manuscript and make possible corrections in the tables and in the entire manuscript. Correction of these values and yield expressions in the DM may affect all calculations in the entire manuscript, including those for energy efficiency indicators.

L188-194. Written too generally. It is necessary to quote the tables to the appropriate rows and values, otherwise it is difficult to track and verify what the authors want to write about. Table citations should be placed in appropriate places, paragraphs of the text. This note applies to the entire manuscript.

L197-198. The sentence is not true, see Table 5, 0N 4489a and 120 N 4493a, so there was no difference, which is very strange and puzzling? Are you sure that at 60 kg/ha N there was an increase in yield, and at 120 kg N there was a decrease to 0 N? The difference between AGB 12030 and R 7037 shows that G = 4993, not 4493. On this basis, I conclude that there are numerical errors in the calculations or other mistakes in the work, which may have serious consequences for further calculations in the entire manuscript and conclusions. Therefore, I recommend that you seriously check all numerical values and calculations in the entire manuscript.

The entire Section “3.3. Energy efficiency for biomass yield and N yield” similar comments as to section 3.3. Please check the calculations, e.g. are the ratios in MJ/kg DM and MJ/kg FM? Please cite tables in appropriate places in the text.

The remaining part of the manuscript was not evaluated due to the serious comments indicated above, which may affect the values, analysis and discussion of the results and conclusions.

Author Response

Dear Reviewer,

Thank you for your comments, which helped us in the improvement of the manuscript.

Followed we are presenting the point-by-point response.

Kind regards

The Authors

The reviewed manuscript presents interesting research. However, I have some very serious remarks and comments. Therefore, at this stage, I propose to reject the manuscript and encourage authors to resubmit it after its verification.

L84 - It is puzzling why these results are being published only now, since the research was carried out in 2010-2011, i.e. over 12 years ago? It needs to be clarified whether anything has changed in this regard over such a long period of time. Have these crops entered commercial production? In addition, is oats not a demanding plant for such good soil? Shouldn't a more demanding crop, such as wheat, be grown here?

Response: Yes, the experiments were carried out in 2010 and 2011 and the energy efficiency analysis was carried out this year based on the agronomy data and technical data.

The technical data (machinery, ernergy input) did not change during this period. Agronomy data are based on the specific site conditions (soil and climate) – and this random influence could have a possible effect. Oat and pea are commercial crops in Austria but normally grown as monocrops. Oat is used for food (oat flakes) and pea as feed for livestock.

Yes, the soil in this experiment is good for cereal cropping. The reasons for using oat were as followed:
a: same harvest time as pea
b: better water use efficiency than wheat

Your recommended crop wheat (spring wheat or durum wheat) could be a good alternative option for oat substitution. Maybe, we will test it at our experimental farm.

L141 – It is necessary to add information and formulas on how the biomass yield, N content and N yield of the grain and residues were calculated. Therefore, after the chapter "2.4. Energy efficiency parameters and energy equivalents", a chapter on the methodology for determining biomass yields, N content and N yields should be added.

Response:

Response: Followed extension to the secion 2.2 was done:

“Plants were harvested manually by cutting on the soil surface at full ripeness on 1.2 m2 on July 21st, 2010, and on July 19th, 2011. The plant samples were divided into grain and residue. Grain and residue samples were first ground to pass through a 1 mm sieve for N determination. N concentration was measured by the Dumas combustion method using an elemental analyzer (vario MACRO cube CNS; Elementar Analysensysteme GmbH, Germany). N concentration data for grain and residues are published in Neugschwandter and Kaul [11].”

[11] Neugschwandtner, R.W.; Kaul, H.-P. Nitrogen uptake, use and utilization efficiency by oat–pea intercrops. Field Crops Research 2015, 179, 113–119, doi:10.1016/j.fcr.2015.04.018.

L147-148. What does it mean that "it was considered as a negligible source of 148 energy" please specify what range or percentage it may be in relation to, for example, other studies that mention it.

Response:

We want to show, that human labor energy input is in comparison to the technical energy input low but it is a little bit confusing. We cancelled this extension.

In the book “Energy in World Agriculture - Editor in Chief B.A. Stout ; Energy in Farm Production – Volume 6 Edited by R.C. Fluck” Elsevier Amsterdam – London – NewYork 1992.
On page 20 is an example: Average energy inputs for producint a hectare of corn in the USA (Pimentel, 1990)

Labor: 25 GJ/ha

Total: 47926 GJ/ha

The entire section “3.2. Crop yields and N yields”. It is necessary to clarify and specify whether crop yields (grain, reidues, AGB) are given in absolutely dry matter (dry matter DM) or at the moisture content that was present during harvesting, i.e. fresh matter (Fresh Matter FM), and then the moisture content of the harvested biomass should be given. It is also necessary to check the calculations and roundings in the whole mansuscript, because e.g. in Table 5, variant 100:0 (GY 5081+RY 6963 = AGB 12044, not 12043 kg/ha). The same is true for N yield (GNY 109.5+RNY 44.9 = AGB 154.4, not 153.8 kg/ha). Therefore, it is necessary to check all calculations in the entire manuscript and make possible corrections in the tables and in the entire manuscript. Correction of these values and yield expressions in the DM may affect all calculations in the entire manuscript, including those for energy efficiency indicators.

Response:

Thank you for showing these mistakes. We checked the Table 5 with the outputs of the statistical programmes.

L188-194. Written too generally. It is necessary to quote the tables to the appropriate rows and values, otherwise it is difficult to track and verify what the authors want to write about. Table citations should be placed in appropriate places, paragraphs of the text. This note applies to the entire manuscript.

Response: Revision was done.

L197-198. The sentence is not true, see Table 5, 0N 4489a and 120 N 4493a, so there was no difference, which is very strange and puzzling? Are you sure that at 60 kg/ha N there was an increase in yield, and at 120 kg N there was a decrease to 0 N? The difference between AGB 12030 and R 7037 shows that G = 4993, not 4493. On this basis, I conclude that there are numerical errors in the calculations or other mistakes in the work, which may have serious consequences for further calculations in the entire manuscript and conclusions. Therefore, I recommend that you seriously check all numerical values and calculations in the entire manuscript.

Response: The numbers were checked with the output of the statistical programme and corrected.

The entire Section “3.3. Energy efficiency for biomass yield and N yield” similar comments as to section 3.3. Please check the calculations, e.g. are the ratios in MJ/kg DM and MJ/kg FM? Please cite tables in appropriate places in the text.

Response: MJ/kg DM

The basis of dry matter (DM) for crop yield and for the relevant energy efficiency parameter (EI and EP) were integrated to the tables.

The remaining part of the manuscript was not evaluated due to the serious comments indicated above, which may affect the values, analysis and discussion of the results and conclusions

Reviewer 2 Report

Dear authors,

I was glad to read your work that shows a system of plant production from an aspect of energy use efficiency.

I hope my comments in the paper will contribute to the quality of your work.

Author Response

Dear Reviewer,

Thank you for your comments, which helped us in the improvement of the manuscript.

Followed we are presenting the point-by-point response.

Kind regards

The Authors

Response to the comments in the pdf manuscript:

Reviewer: Please do not use the commercial name of the plant protection product, only state the active substance.

Response: Decis was subtituted by “active substance: deltametrin”.

Reviewer: It is necessary to write how you determined the yield and nitrogen content.

How sampling and analysis of aboveground biomass was carried

and also plant residues?

Response: Followed extension to the secion 2.2 was done:

“The plant samples were divided into grain and residue. Grain and residue samples were first ground to pass through a 1 mm sieve for N determination. N concentration was measured by the Dumas combustion method using an elemental analyzer (vario MACRO cube CNS; Elementar Analysensysteme GmbH, Germany). N concentration data for grain and residues are published in Neugschwandter and Kaul [11].”

Reviewer: It is necessary to expand the discussion in this part and support it with references.

In intercropping systems it is important to highlight plant species.

Legumes and grasses naturally have different yields, but grasses usually serve as carriers of legumes in the initial stages of growth and development.

The uptake of nutrients, especially nitrogen, is different for legumes and grasses. Nitrogen content in legumes is not only the result of nitrogen fertilization, but also of symbiotic relationships with nitrofixing bacteria.

Due to the competition between grasses and legumes and in the initial stages of development, the formation of symbiosis with nodule bacteria on the roots of legumes can often be even more pronounce.

It is also important to mention LER (Land Equivalent Ratio) that is an important indicator of intercropping efficiency.

Response:  We agree with you. These mentioned aspects including LER are already published in the followed paper, which are based on the same field experiment.

Neugschwandtner, R.W.; Kaul, H.-P. Sowing ratio and N fertilization affect yield and yield components of oat and pea in intercrops. Field Crops Research 2014, 155, 159–163, doi:10.1016/j.fcr.2013.09.010.

Neugschwandtner, R.W.; Kaul, H.-P. Concentrations and uptake of macronutrients by oat and pea in intercrops in response to N fertilization and sowing ratio. Archives of Agronomy and Soil Science 2016, 62, 1236–1249, doi:10.1080/03650340.2016.1147648.

The focus in this present manuscript is energy efficiency.

Reviewer 3 Report

This manuscript reports on a valuable two-year field study, with practical implications for farmers and policy makers, to determine the Energy budget of intercropping systems, as affected by Nitrogen fertilization. The study follows a standard and straightforward methodology, is very well written, and is well organized.

On the methodology, provide additional background information on the planting of the barley crop, prior to beginning the experiment (see comment below for L 112).

Also on the methodology I suggest that additional clarifications be provided with respect to the initial N status of the experimental plots. Was the pea intercrop inoculated with Rhizobia, or was effective nodulation observed in the field from native/existing soil Rhizobial populations? What percentage of the N budget in the intercropping system is provided by N fixation, in fields with effective inoculation? Also, what percentage of the N budget for oats is provided by the intercropped pea N fixation, if any?

 In the Discussion, provide some context and analysis on the N status of the soil, prior to experiment initiation, whether the initial N soil levels had an effect on the experimental results, and if so, whether these initial soil N levels had an effect on the overall Energy budget analysis. Overall, would the energy budget analysis be different in fields where N biological fixation played a more prominent role as part of the N system budget (I imagine that the zero N controls provide insight on this respect)?

I suggest that the authors double check the presentation of the results in terms of their description of statistical significance between treatments (see comments below for L244 and L249, as examples).

Comments on the text include,

L 112-113, May want to indicate whether the previous barley crop was unfertilized, and whether the barley residues were removed from the field after harvest, or soil incorporated.

L 244, Please double check this statement. Table 8 shows no significant differences between intercropping ratios, except for lower values with 100:0.

L 249-250, Please double check the values and statistics. Table 8 shows no statistical difference among treatments, except for 0:100 (pure peas).

L 249-251, Please note that according to Table 8, the Grain-N-yield values were statistically equivalent for the 100:0, 50:50, and 25:75 intercrops; and that 100:0, 50:50, and 25:75 are also statistically equivalent. Thus Grain-N-yield values were higher than pure pea when oat was planted, but not at increasing ratios, i.e. values were statistically equivalent with oat planting ratios of 100:0 and 25:75.

Additional minor edit suggestions are included in the attached copy of the manuscript.

/////

Author Response

Dear Reviewer,

Thank you for your comments, which helped us in the improvement of the manuscript.

Followed we are presenting the point-by-point response.

Kind regards

The Authors

This manuscript reports on a valuable two-year field study, with practical implications for farmers and policy makers, to determine the Energy budget of intercropping systems, as affected by Nitrogen fertilization. The study follows a standard and straightforward methodology, is very well written, and is well organized.

On the methodology, provide additional background information on the planting of the barley crop, prior to beginning the experiment (see comment below for L 112).

Response: Followed sentences were integrated:

“Winter barley and spring barley were fertilized with 100 kg N ha−1. Barley residues were soil incorporated.”

Also on the methodology I suggest that additional clarifications be provided with respect to the initial N status of the experimental plots. Was the pea intercrop inoculated with Rhizobia, or was effective nodulation observed in the field from native/existing soil Rhizobial populations? What percentage of the N budget in the intercropping system is provided by N fixation, in fields with effective inoculation? Also, what percentage of the N budget for oats is provided by the intercropped pea N fixation, if any?

Response: Pea seed was not inoculated with Rhizobia.

In the Section 2.2. followed sententence was integrated:

"Pea seed was not rhizobial inoculated."

In the Discussion, provide some context and analysis on the N status of the soil, prior to experiment initiation, whether the initial N soil levels had an effect on the experimental results, and if so, whether these initial soil N levels had an effect on the overall Energy budget analysis. Overall, would the energy budget analysis be different in fields where N biological fixation played a more prominent role as part of the N system budget (I imagine that the zero N controls provide insight on this respect)?

Response: A very good question and difficult to answer:

We added two sentences in the discussion:

“Depending on the soil water availability, the soil mineral N from different sources (organic and mineral fertilizer, biological nitrogen fixation, humus, N deposition) is mainly responsible for the biomass production and energy output. It is supposed, that the energy efficiency indicators will be affected positively, if the soil mineral N is not delivered by the energy-intensive mineral N fertilizer.”

I suggest that the authors double check the presentation of the results in terms of their description of statistical significance between treatments (see comments below for L244 and L249, as examples).

Comments on the text include,

Response: Check was done. Mistake was corrected: 55:50 => 75:25

L 112-113, May want to indicate whether the previous barley crop was unfertilized, and whether the barley residues were removed from the field after harvest, or soil incorporated.

Response: Followed sentence was integrated:

"Winter and spring barley were fertilized with 100 kg N ha−1. Barley residues were soil incorporated."

L 244, Please double check this statement. Table 8 shows no significant differences between intercropping ratios, except for lower values with 100:0.

 Response: The sentence was linked with Table 9, because there was an interaction between sowing ratio and N fertilization. There was also found an interaction between N fertilization and year (Table 11).

We make it clear with intergration of “Table 9” and “Table 11” after the relvant sentence.

L 249-250, Please double check the values and statistics. Table 8 shows no statistical difference among treatments, except for 0:100 (pure peas).

Response: The linkages to the tables were corrected.

L 249-251, Please note that according to Table 8, the Grain-N-yield values were statistically equivalent for the 100:0, 50:50, and 25:75 intercrops; and that 100:0, 50:50, and 25:75 are also statistically equivalent. Thus Grain-N-yield values were higher than pure pea when oat was planted, but not at increasing ratios, i.e. values were statistically equivalent with oat planting ratios of 100:0 and 25:75.

Response: But there were significant interactions between sowing ration x fertilization as well fertilization x year for the variables EIGRAIN-YIELD and EIAGB_N-YIELD. The statistical results for the interactions are shown in Table 9 and 11. The linkages were integrated.

Additional minor edit suggestions are included in the attached copy of the manuscript.

Round 2

Reviewer 1 Report

The manuscript has been corrected